# Different Roles of Water Vapor Transport and Cold Advection in the Intensive Snowfall Events over North China and the Yangtze River Valley

Zhixing Xie [1] and Bo Sun [1,2,*] 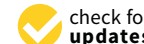

[1] Collaborative Innovation Center on Forecast and Evaluation of Meteorological Disasters/Key Laboratory of Meteorological Disasters, Ministry of Education/Joint International Research Laboratory of Climate and Environment Change, Nanjing University of Information Science and Technology, Nanjing 210044, China

[2] Nansen-Zhu International Research Centre, Institute of Atmospheric Physics, Chinese Academy of Sciences, Beijing 100029, China

* Correspondence: sunb@nuist.edu.cn

**Abstract:** Intensive snowfall events (ISEs) have a profound impact on the society and economy in China during winter. Considering that the interaction between northerly cold advection and southerly water vapor transport (WVT) is generally an essential condition for the occurrence of ISEs in eastern China, this study investigates the different roles of anomalous southerly WVT and northerly cold advection during the ISEs in the North China (NC) and Yangtze River valley (YRV) regions based on a composite analysis of seventy ISE cases in NC and forty ISE cases in the YRV region from 1961 to 2014. The results indicate that the ISEs in NC are mainly associated with a significant pre-conditioning of water vapor over NC induced by southerly WVT anomalies over eastern China, whereas the ISEs in the YRV region are mainly associated with a strengthened Siberian High (SH) and strong northerly cold advection invading the YRV region. These results suggest a dominant role of anomalous southerly WVT in triggering the ISEs in NC and a dominant role of northerly cold advection in triggering the ISEs in the YRV region. The different roles of anomalous southerly WVT and northerly cold advection in the ISEs over the NC and YRV regions are largely attributed to the different winter climate in the NC and YRV regions—during winter, the NC (YRV) region is dominated by cold and dry (relatively warm and moist) air flow and hence southerly WVT (northerly cold advection) is the key factor for triggering the ISEs in NC (the YRV region).

**Keywords:** intensive snowfall events; water vapor transport; cold advection; North China; Yangtze River valley

## 1. Introduction

During winter, intensive snowfall events (ISEs) are one of the most influential synoptic disasters, which can widely impact the agriculture, transportation and social activities [1–3]. Many studies have been aroused to investigate the variability of ISEs and the dynamic processes associated with the ISEs around the world [4–17]. Particularly for China, in January 2008, persistent snowy and icy weathers occurred in a wide area of southern China, causing enormous economic losses in seven provinces including Anhui, Jiangxi, Hubei, Hunan, Guangxi, Sichuan and Guizhou [18–20]; in January 2018, two ISEs occurred in the Yangtze River valley (YRV) region and adjacent regions, which resulted in traffic jams and casualties in a wide area including the Anhui, Jiangsu, Jiangxi, Shanxi, Henan and Hubei provinces [3]. Over past decades, the extreme precipitation, including heavy snowfall, has been increased during winter in most areas of China [21], which warrants more public concerns about the ISE in China.

The anomalous atmospheric circulation and associated influential factors (e.g., water vapor, cold advection, regional convection, topography) of ISEs in different regions of China have been studied [3,19–30]. It has been learnt that most of the ISEs in China are closely associated with an interaction between southerly water vapor transport (WVT) and northerly cold advection [3,22]. The cold advection over China is generally affected by the Siberian High (SH), which is associated with a surface anticyclone over the Eurasian continent and which has a non-stationary relationship with the Arctic Oscillation [17,31,32]. Specifically, Sun et al. [24] focused on the ISE in northeastern China on 3–5 March 2007, arguing that a main causal factor of this ISE was the robust anomalous southerly WVT. Wang et al. [18] analyzed several intensive snowfall cases in southern China in January 2008, claiming that the major factor triggering the snowfall was the low level southeasterly jet, which carried abundant moisture from the low-latitudinal oceans, and which interacted with an anomalous northerly cold airflow associated with a strengthened SH. Moreover, based on a composite analysis on the widespread snowfall events in northeastern China during 1979–2009, Sun and Wang [22] suggested that the southerly WVT originating from eastern China, the Yellow Sea and the Japan Sea played an important role in supplying water vapor for triggering and sustaining the widespread snowfall events in northeastern China, which is companied by an anomalous strong SH and a negative phase of Arctic Oscillation. Recently, significant southerly WVT anomalies and northerly cold advection were observed during the ISEs in the YRV region in January 2018, suggesting important roles of southerly WVT and northerly cold advection in inducing the ISEs in the YRV region [3]. In addition, the middle-to-upper tropospheric processes may also exert an impact on the extreme precipitation including ISEs in China [33–35].

Considering that the northern China is dominated by northerly cold air flow during winter [22], while the southern China is generally dominated by relatively warm and moist air flow during winter [3], the roles of southerly WVT and northerly cold advection in the ISEs in different regions of China are possibly different. Moreover, it should be noted that the ISEs in different regions may be associated with different water vapor sources and WVT paths [22,25]— in northeastern China, the ISEs are mainly associated with cyclonic WVT anomalies over northeastern China, where anomalous southerly and southeasterly WVT carry moisture from eastern China and the adjacent seas into northeastern China [22]; whereas, the two ISEs in the central eastern China occurring in January 2018 are associated with anti-cyclonic WVT anomalies over the western North Pacific and southern China, where the anomalous southwesterly WVT carry moisture from the Indochina Peninsula and South China Sea into the central eastern China [3]. Previous studies mostly focused on the variability and mechanism associated with the ISEs in northern China [22,24]. The different roles of WVT and cold advection in the ISEs over different regions have been less understood.

Thus, this study aims to investigate the different roles of WVT and cold advection during the ISEs in North China (NC) and the YRV region, focusing on the ISEs in winter (December-January-February, DJF) during 1961–2014.

The outline of this paper is as follows. Section 2 describes the data and method used in this study. In Section 3, the climatology of influential factors of ISEs (water vapor and northerly wind) over East Asia during winter is examined. The synoptic evolutions of WVT and cold advection associated with the ISEs in the NC and YRV regions are investigated and discussed in Section 4. Section 5 provides conclusion and discussion.

## 2. Data and Methods

### 2.1. Data

The 12-h accumulative precipitation data and daily mean surface temperature data are derived from the station data archived in the China Meteorological Administration (http://data.cma.cn/), which span from 1951 to present and contain data from over 800 gauge stations in China. The National Centers for Environmental Prediction (NCEP) reanalysis data (https://www.esrl.noaa.gov/psd/data/

gridded/data.ncep.reanalysis.html) used in this study include monthly and four-times daily data of surface wind, surface pressure, precipitable water, sea level pressure (SLP) and wind as well as specific humidity at eight pressure levels (1000, 925, 850, 700, 600, 500, 400, 300-hPa), which have a resolution of 2.5° × 2.5°.

Considering that the station data use Beijing Time (BT) while the NCEP reanalysis data use the coordinated universal time (UTC), the time axis of the station data was transformed into the UTC before all computations. Specifically, the time span 2000–0800 BT and 0800–2000 BT for the 12-h accumulative precipitation correspond to the time span 1200–2400 UTC and 0000–1200 UTC, respectively; the time span 2000–2000 BT for the daily mean surface temperature corresponds to the time span 1200–1200 UTC.

*2.2. Methods*

Focusing on NC (35°–42.5° N, 110°–122.5° N) and the YRV region (27.5°–35° N, 110°–122.5° N), precipitation and temperature data from 74 stations in NC and 76 stations in the YRV region were selected to perform the analysis (Figure 1), focusing on the period of 1961–2014.

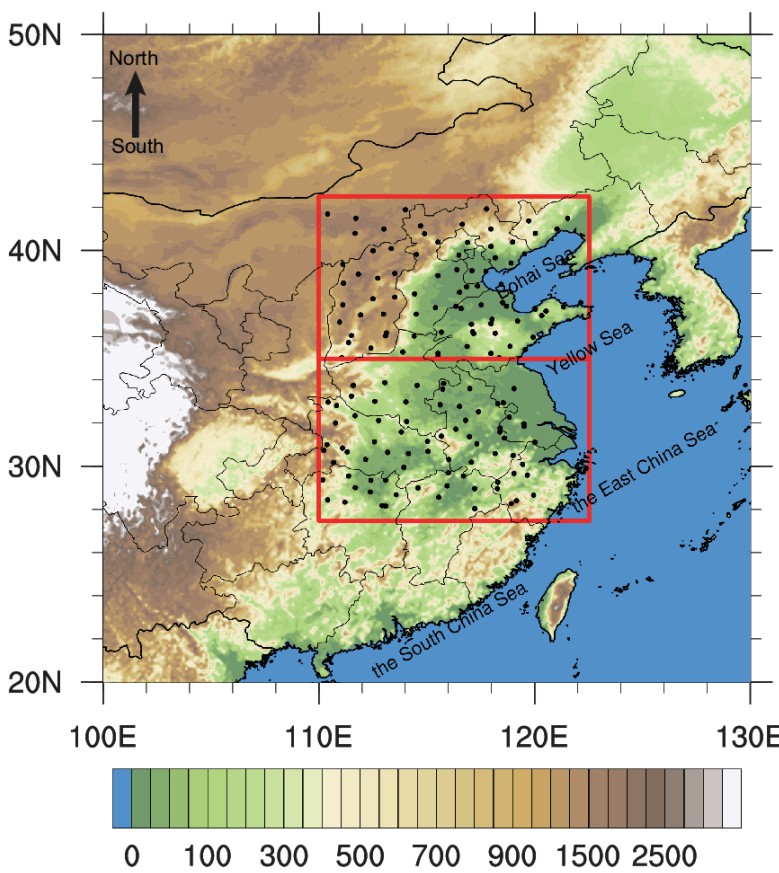

**Figure 1.** Spatial distribution of 74 gauge stations (solid dots) in NC and 76 gauge stations (solid dots) in the Yangtze River Valley (YRV) region and the topography (colors, m). The topography data were derived from the Global Land One-km Base Elevation Project database (http://www.ngdc.noaa.gov/mgg/topo/globe.html).

Previous studies have proposed several criteria for defining the ISEs in different regions of China based on precipitation and temperature [21,22]. In this study, the ISEs in the target regions during 1961–2014 are identified according to the below criteria— (1) the 12-h accumulative precipitation should be larger than 0.1 mm and the daily mean surface temperature should be below the freezing point; (2) more than 30% of the 74 (76) stations in NC (the YRV region) should meet the above condition;

(3) the events meeting the above criteria are ranked in terms of the total precipitation amount of stations in the target region, where the top 30% events are identified as ISEs in the NC and YRV regions. In the above criterion (3), the total precipitation amount is calculated by the sum of 12-h accumulative precipitation of the stations meeting criterion (1). Table 1 shows the 70 ISEs in NC and 40 ISEs in the YVR region during 1961–2014 identified based on the above criteria. Most of the ISEs in the YRV region occurred during 1200–2400 UTC (2000–0800 BT). This phenomenon may be mainly because that the temperature during night (2000–0800 BT) is lower than during day (0800-2000 BJ), which is a better condition for the occurrence of ISEs. Following Sun and Wang [22], the time axis in the composite analysis is set based on the beginning time of ISEs, where the 0 time step represents the beginning time of ISEs, the −12 time step represents the 12 h before the breakout of ISEs, the 12 time step represents the 12 h after the breakout of ISEs and so on.

**Table 1.** The 70 intensive snowfall event (ISE) cases in North China (NC) and 40 ISE cases in the YRV region used for the composite analysis. The cases are sorted by the total precipitation amount of stations in the target regions.

| NC | | | YRV | | |
|---|---|---|---|---|---|
| Total Precipitation Amount (mm) | Date | UTC | Total Precipitation Amount (mm) | Date | UTC |
| 340.4 | 1976.2.14 | 1200–2400 | 615.9 | 1984.1.17 | 1200–2400 |
| 323.5 | 1979.2.21 | 1200–2400 | 597.6 | 2010.2.10 | 1200–2400 |
| 240.9 | 1990.1.27 | 1200–2400 | 438 | 1972.2.4 | 1200–2400 |
| 232.2 | 2001.1.6 | 0000–1200 | 411.4 | 1991.12.26 | 0000–1200 |
| 175.9 | 1973.1.22 | 1200–2400 | 381 | 1969.1.10 | 1200–2400 |
| 154.9 | 2010.2.28 | 0000–1200 | 358.8 | 1966.2.21 | 1200–2400 |
| 151.6 | 1964.2.14 | 1200–2400 | 356.4 | 1965.12.15 | 1200–2400 |
| 150.1 | 1972.1.30 | 0000–1200 | 342.4 | 1969.1.27 | 1200–2400 |
| 147.2 | 1986.12.26 | 1200–2400 | 300.8 | 1974.2.23 | 1200–2400 |
| 147.1 | 1972.1.23 | 0000–1200 | 287.9 | 1964.2.16 | 0000–1200 |
| 145.7 | 1979.1.28 | 0000–1200 | 279.9 | 1989.2.22 | 0000–1200 |
| 139.8 | 1970.2.23 | 1200–2400 | 279.9 | 1964.2.7 | 1200–2400 |
| 136.7 | 2012.12.28 | 1200–2400 | 277.2 | 1990.1.29 | 1200–2400 |
| 134.7 | 1985.12.6 | 1200–2400 | 277 | 1997.1.22 | 1200–2400 |
| 125.1 | 1969.2.13 | 0000–1200 | 274.5 | 1969.2.2 | 1200–2400 |
| 122.9 | 2001.2.22 | 1200–2400 | 257.3 | 2008.1.27 | 0000–1200 |
| 122.7 | 1966.2.20 | 1200–2400 | 238.7 | 1966.12.24 | 1200–2400 |
| 120.4 | 1994.2.10 | 1200–2400 | 235.8 | 2011.1.19 | 1200–2400 |
| 119.4 | 1971.12.23 | 1200–2400 | 231.4 | 1994.2.23 | 1200–2400 |
| 115.8 | 1964.1.10 | 0000–1200 | 226.9 | 2008.2.1 | 0000–1200 |
| 115 | 2012.12.20 | 1200–2400 | 223.3 | 1979.1.30 | 0000–1200 |
| 114.2 | 1963.12.9 | 0000–1200 | 223.2 | 1996.2.16 | 1200–2400 |
| 113.5 | 1981.1.22 | 1200–2400 | 219 | 1972.2.2 | 1200–2400 |
| 109.3 | 1978.2.7 | 1200–2400 | 212.6 | 1993.1.13 | 1200–2400 |
| 108.9 | 1974.2.4 | 0000–1200 | 201.2 | 1994.1.17 | 1200–2400 |
| 108.8 | 1962.2.9 | 0000–1200 | 184.7 | 1989.1.12 | 1200–2400 |
| 105.5 | 2006.2.6 | 0000–1200 | 183.7 | 2003.2.10 | 1200–2400 |
| 105.1 | 2010.1.3 | 0000–1200 | 178.7 | 1983.12.28 | 1200–2400 |
| 103.5 | 1996.2.16 | 0000–1200 | 173.2 | 1985.12.9 | 0000–1200 |
| 103.2 | 1989.1.5 | 1200–2400 | 170.1 | 1979.1.11 | 0000–1200 |
| 102.9 | 1964.2.5 | 1200–2400 | 169.2 | 2008.1.14 | 1200–2400 |
| 99 | 1993.1.8 | 0000–1200 | 166 | 1977.1.28 | 1200–2400 |
| 98.9 | 2009.2.18 | 1200–2400 | 158.2 | 1964.2.22 | 1200–2400 |
| 98 | 1981.2.19 | 0000–1200 | 156.3 | 1985.2.16 | 1200–2400 |
| 97 | 1984.12.15 | 0000–1200 | 156.1 | 2008.1.18 | 1200–2400 |

**Table 1.** *Cont.*

| NC | | | YRV | | |
|---|---|---|---|---|---|
| Total Precipitation Amount (mm) | Date | UTC | Total Precipitation Amount (mm) | Date | UTC |
| 96.7 | 1971.1.17 | 1200–2400 | 155.9 | 2000.1.24 | 0000–1200 |
| 95.9 | 1990.1.29 | 1200–2400 | 149.8 | 1967.2.9 | 1200–2400 |
| 95.9 | 2000.1.21 | 1200–2400 | 126.9 | 1988.2.16 | 0000–1200 |
| 94.2 | 1975.2.3 | 0000–1200 | 126.8 | 1970.1.4 | 0000–1200 |
| 93.5 | 1973.2.2 | 0000–1200 | 126.5 | 1983.1.11 | 0000–1200 |
| 92 | 1971.12.21 | 1200–2400 | | | |
| 90.3 | 1969.1.27 | 1200–2400 | | | |
| 89.7 | 2012.12.13 | 0000–1200 | | | |
| 88.9 | 1976.2.17 | 0000–1200 | | | |
| 87.7 | 2006.2.27 | 0000–1200 | | | |
| 86.4 | 1987.2.18 | 1200–2400 | | | |
| 84.3 | 1983.1.31 | 1200–2400 | | | |
| 84.2 | 2002.12.22 | 0000–1200 | | | |
| 83.7 | 1972.2.15 | 0000–1200 | | | |
| 82.4 | 1991.12.22 | 0000–1200 | | | |
| 81.7 | 2005.2.14 | 1200–2400 | | | |
| 81.4 | 1967.1.26 | 1200–2400 | | | |
| 80.4 | 1969.2.15 | 0000–1200 | | | |
| 80.4 | 1986.2.16 | 1200–2400 | | | |
| 80.3 | 2011.2.25 | 1200–2400 | | | |
| 79.4 | 2002.12.6 | 1200–2400 | | | |
| 79.3 | 1997.1.4 | 0000–1200 | | | |
| 78.8 | 1979.12.18 | 0000–1200 | | | |
| 76.8 | 2008.2.24 | 1200–2400 | | | |
| 76.4 | 1982.2.3 | 0000–1200 | | | |
| 76.1 | 2010.2.10 | 0000–1200 | | | |
| 74.8 | 2001.2.5 | 0000–1200 | | | |
| 72.9 | 2013.2.3 | 0000–1200 | | | |
| 72.9 | 2001.1.23 | 1200–2400 | | | |
| 70.4 | 1989.12.18 | 1200–2400 | | | |
| 69.5 | 2000.1.5 | 0000–1200 | | | |
| 67.6 | 1962.2.24 | 0000–1200 | | | |
| 63.5 | 1994.12.18 | 1200–2400 | | | |
| 62.7 | 1981.12.17 | 1200–2400 | | | |
| 62.2 | 2014.2.4 | 1200–2400 | | | |

The vertically integrated WVT and the water vapor flux budget over the target regions are calculated based on the method developed by Simmonds [36] and Sun et al. [37], using the zonal and meridional winds and specific humidity at eight pressure levels (1000, 925, 850, 700, 600, 500, 400, 300-hPa). Following Wu and Wang [11], the SH index is calculated by the SLP averaged over the region within 40°–60° N and 80°–120° E. The station data of surface temperature and precipitation are interpolated into a gridded data with a resolution of 0.5° × 0.5°.

## 3. Climatology of Water Vapor Transport, Cold Advection and Precipitation over Eastern China during Winter

The winter climate over East Asia is modulated by the SH, where a strengthened SH generally induces a strong East Asian winter monsoon and vice versa [10,30,31,38–41]. As shown in Figure 2a, the SH is generally characterized by large SLP over Mongolia, which induces divergent surface wind over East Asia, resulting in northerly surface wind over eastern China. The northerly surface wind associated with the SH is a key factor for the cold advection invading eastern China during winter.

Figure 2b shows the climatology of surface meridional wind over East Asia. It can be seen that the NC and YRV regions are both dominated by northerly surface wind during winter, where the northerly surface wind over NC is mostly larger than the northerly surface wind over the YRV region, suggesting more frequent cold advection over NC than the YRV region during winter. In addition, the surface temperature over NC is mostly near or below 0 °C during winter, whereas the surface temperature over the YRV region is noticeably above 0 °C (Figure 2e). The association between the infrequent northerly cold advection and the relatively small quantity of ISEs over the YRV region (Table 2) implies a key role of the northerly cold advection for triggering the ISE in the YRV region during winter.

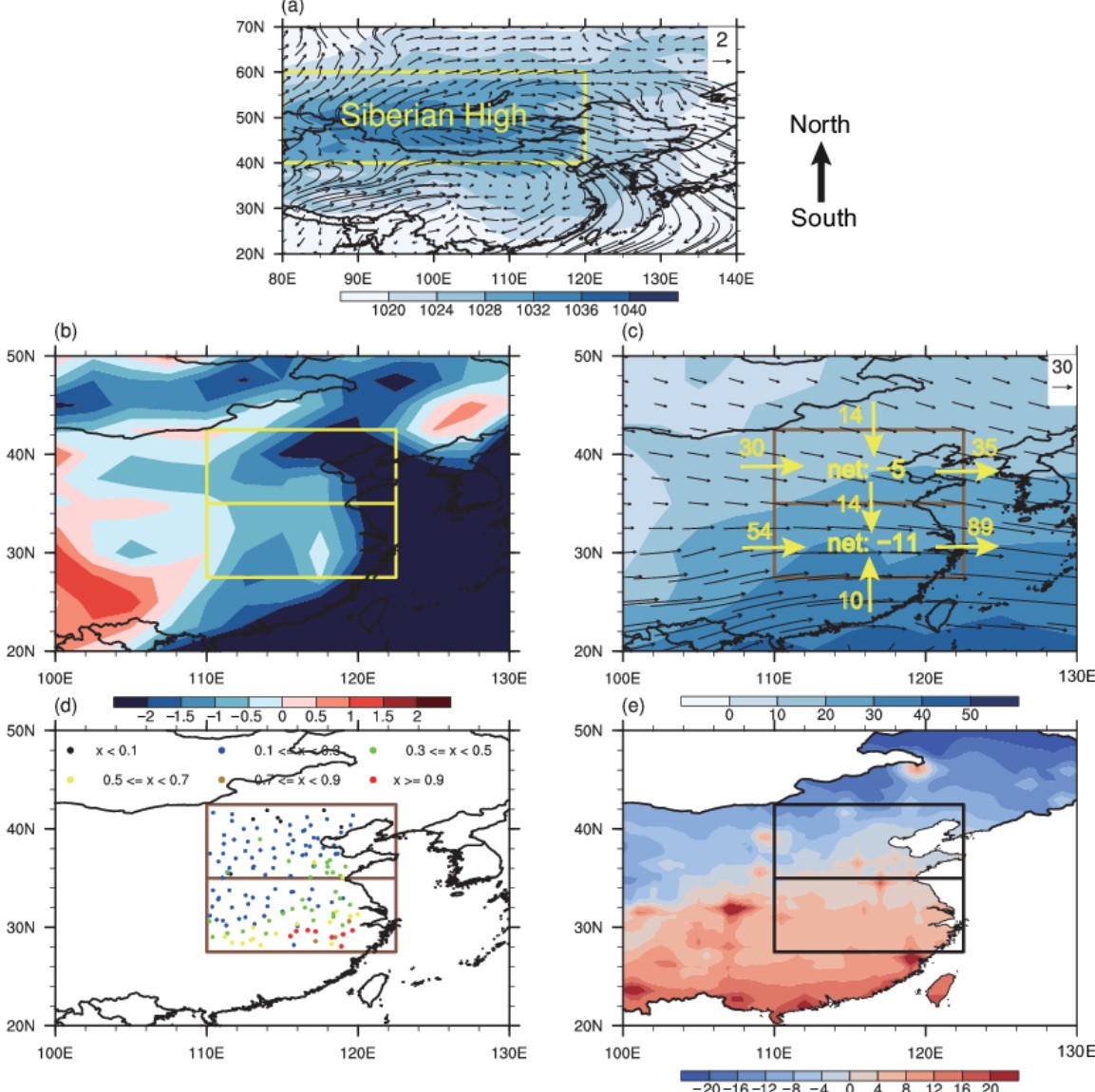

**Figure 2.** Winter (DJF) climatology of (**a**) sea level pressure (SLP) (colors, hPa) and surface wind (vectors, m/s), (**b**) surface meridional wind (colors, m/s), (**c**) vertically integrated water vapor transport (WVT) (vectors, kg/(m·s)), total precipitable water (colors, kg/m$^2$) and associated water vapor flux budget over NC and the YRV region (red vector, $10^6$ kg/s), (**d**) gauged precipitation (colored dots, mm/day) and (**e**) surface temperature (colors, °C). The rectangle in (**a**) denotes the region where the SLP is used for calculating the Siberian High (SH) index. The rectangles in (**b–e**) denote the domains of NC and the YRV region.

The climatology of WVT over NC and the YRV region during winter also exhibits different features, where northwesterly WVT prevails over NC and westerly WVT prevails over the YRV region

(Figure 2c). Specifically, the water vapor flux budget over the two target regions indicates a negative net water vapor flux for both regions, which is mainly caused by the large water vapor outflux through the eastern boundaries of the two regions (Figure 2c). Notwithstanding the relatively large negative net water vapor flux over the YRV region in comparison to NC, the precipitable water over the YRV region is noticeably larger than that over NC during winter, partially due to the southwesterly WVT over southern China carrying warm and moist air into the YRV region and partially due to the northwesterly WVT over NC carrying cold and dry air into NC (Figure 2c). Correspondingly, the winter precipitation over eastern China exhibits a southeast-northwest gradient, with relatively large precipitation in the southeastern part of eastern China and relatively small precipitation in the northwest part of eastern China (Figure 2d), which is consistent with the spatial distribution of precipitable water, suggesting an important influence of water vapor on the winter precipitation in the NC and YRV regions.

**Table 2.** Percentage of ΔWVT, ΔO, ΔP during water vapor pre-conditioning stage (time step −24 to 0) and during the IS stage (time step 0 to 48) accounting for the total precipitation during the whole period (time step −24 to 48) for the ISEs in NC and the YRV region.

| | NC | | | YRV | | |
|---|---|---|---|---|---|---|
| | **ΔWVT** | **ΔO** | **ΔP** | **ΔWVT** | **ΔO** | **ΔP** |
| **Water Vapor Pre-Conditoning Stage** | 73% | −57% | 16% | 16% | 10% | 26% |
| **IS Stage** | −10% | 94% | 84% | −38% | 112% | 74% |

Overall, the above results indicate different climatic conditions over NC and the YRV region during winter regarding WVT and cold advection. For NC, cold advection frequently invades this region during winter (Figure 2a,b), whereas southerly WVT carrying warm and humid air can hardly reach this region (Figure 2c). The southerly WVT may be a key factor for triggering the ISEs in NC. By contrast, for the YRV region, relatively warm and humid air dominates this region during winter (Figure 2c), which provides a favorable water vapor condition for the winter precipitation and ISE in the YRV region; whereas, less northerly cold advection invades this region during winter (Figure 2a,b). The northerly cold advection may be a key factor for triggering the ISEs in the YRV region.

## 4. Synoptic Features of Intensive Snowfall Events in North China and the Yangtze River Valley Region

### 4.1. Synoptic Evolution of Water Vapor Transport

Figure 3 shows the precipitation during the ISEs in the NC and YRV regions. For both regions, the 12-h accumulative precipitation reaches the peak value 12 h after the breakout of ISEs and decreases gradually afterwards, with a relatively small precipitation amount in NC and a relatively large precipitation amount in the YRV region. The spatial distributions of precipitation for the ISEs in NC and for the ISEs in the YRV region are distinct. Note that there is also precipitation observed in the area surrounding the target regions, which may be because that the ISEs in eastern China are generally associated with large-scale synoptic systems that could result in precipitation over a large area. The NCEP precipitation data can well reproduce the evolution of precipitation during the ISEs in the target regions (figure not shown), indicating a good capability of the NCEP data for analyzing the ISEs in eastern China. Correspondingly, Figures 4 and 5 show the synoptic evolutions of vertically integrated WVT anomalies with respect to winter climatology from 18 h before the breakout of ISEs to 24 h after the breakout of ISEs for NC and the YRV region, respectively. For the ISEs in NC, southerly WVT anomalies dominated eastern China in the 18 h before the breakout of ISEs, indicating moisture originating from eastern China, the South China Sea and western North Pacific (Figure 4a–d). These southerly WVT anomalies resulted in significant water vapor convergence over NC (Figure 4a–d), which contributed to increased water vapor. Afterwards, in the 24 h after the breakout of ISEs, the southerly WVT anomalies over NC gradually became southwesterly WVT anomalies (Figure 4e–h), indicating weakened southerly WVT anomalies over NC; correspondingly, the

convergence of water vapor gradually decreased over NC and a gradually increased divergence in water vapor occurred over NC (Figure 4e–h), which may reduce the water vapor supply for the precipitation in NC. Similarly, for the ISEs in the YRV region, southerly WVT anomalies dominated the southern China and the YRV region in the 18 h before the breakout of ISEs, resulting in a significant convergence in water vapor and contributing to increased water vapor over the YVR region (Figure 5a–d); whereas, in the 24 h after the breakout of ISEs, the southerly WVT anomalies over the YRV region gradually became southwesterly WVT anomalies and a divergence of water vapor occurred over the YRV region (Figure 5e–h), which induced a decrease in water vapor supply for the precipitation in the YRV region. A main difference in the synoptic evolution of WVT anomalies for the ISEs in NC and the YRV region is that the convergence of water vapor over NC in the 24 h before the breakout of ISEs (Figure 4a–d) is notably larger than the convergence of water vapor over the YRV region (Figure 5a–d), implying a more close connection between the ISEs in NC and water vapor convergence than the connection between the ISEs in the YRV region and water vapor convergence.

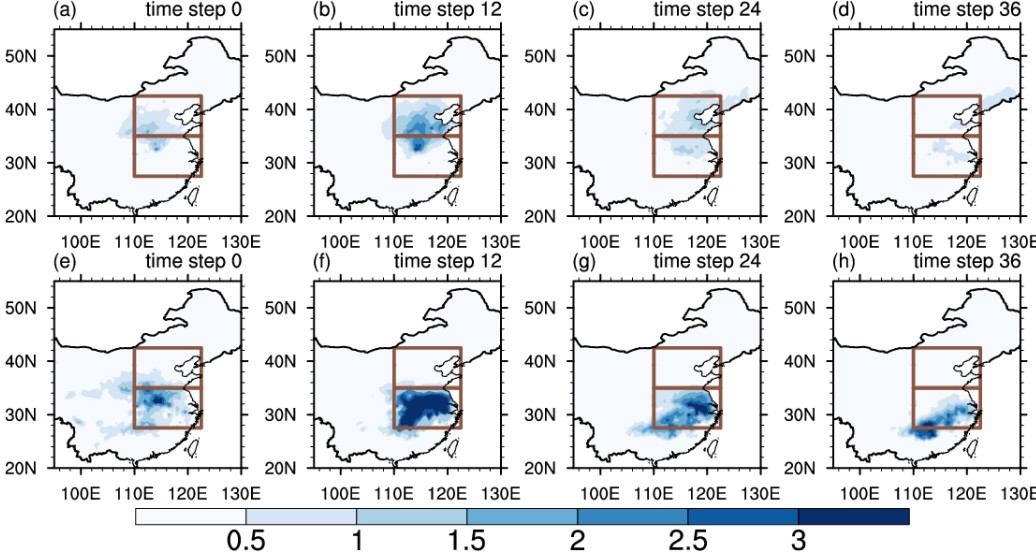

**Figure 3.** Evolution of 12-h accumulative precipitation (colors, mm) from time step 0 to 36 for ISEs in (**a–d**) NC and (**e–f**) the YRV region. The 12-h accumulative precipitation with surface temperature above the freezing point has been masked.

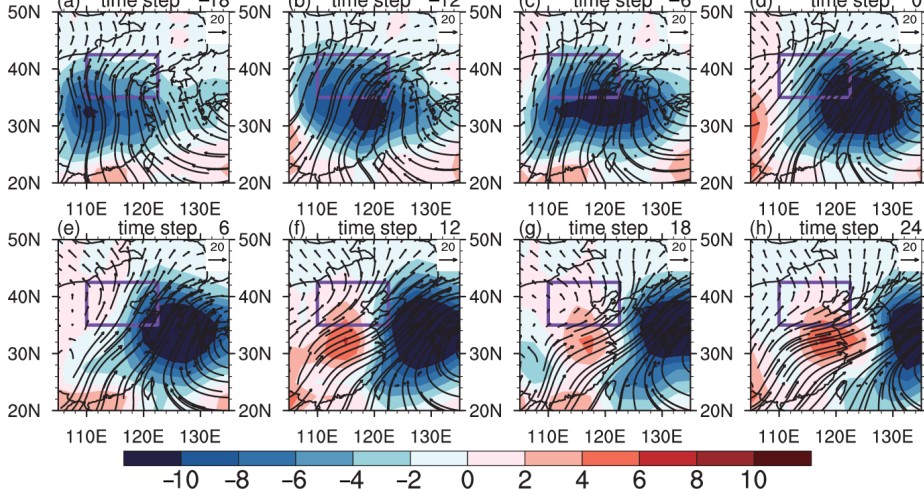

**Figure 4.** Evolution of vertically integrated WVT anomalies (vectors, kg/(m·s) with respect to winter climatology and the associated divergence (colors, $10^{-5}$ kg/(m$^2$·s)) at time steps (**a**) −18, (**b**) −12, (**c**) −6, (**d**) 0, (**e**) 6, (**f**) 12, (**g**) 18, and (**h**) 24 for ISEs in NC.

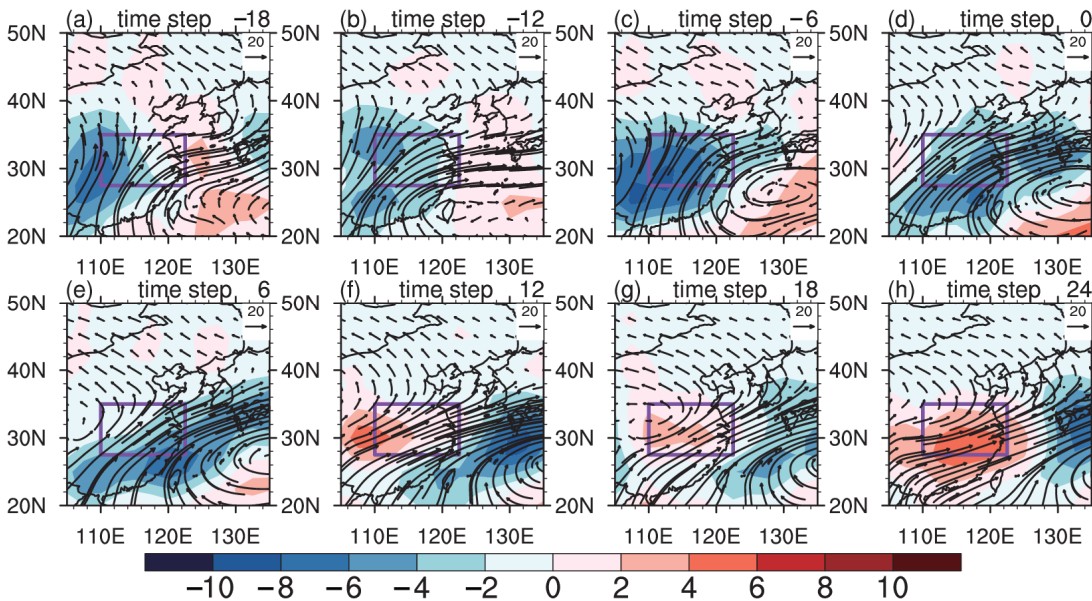

**Figure 5.** Evolution of vertically integrated WVT anomalies (vectors, kg/(m·s) with respect to winter climatology and the associated divergence (colors, $10^{-5}$ kg/(m²·s)) at time steps (**a**) −18, (**b**) −12, (**c**) −6, (**d**) 0, (**e**) 6, (**f**) 12, (**g**) 18, and (**h**) 24 for ISEs in the YRV region.

To further compare the synoptic evolution in precipitation and water vapor flux budget over the two target regions during the ISEs, Figure 6 shows the evolution of areal mean 12-h cumulative precipitation and net water vapor flux over the two target regions. The precipitation in the two regions began notably increasing after the beginning of the ISEs, peaking in the 12 h after the breakout of ISEs and decreasing gradually afterwards (Figure 6a,c,e). In contrast, the net water vapor flux over the two regions began increasing in the 24 h before the breakout of ISEs, reaching the peak in the 6–12 h before the breakout of ISEs and decreasing gradually (Figure 6b,d,f). The net water vapor flux over the two regions fell towards a value below the level of winter climatology in the 12–36 h after the breakout of ISEs. This lead-lag relationship between the synoptic evolution of water vapor flux budget and precipitation indicates an important role of the water vapor pre-conditioning over the two target regions for the ISEs, which is similar to the lead-lag relationship between the water vapor flux budget and the precipitation during the ISEs in northeastern China [22]. What should be noted is that the precipitation in the YRV region is larger than the precipitation in NC during the ISEs (Figure 6e), whereas the anomalies of net water vapor flux over NC is larger than that over the YRV region (Figure 6f), indicating a stronger water vapor pre-conditioning via large-scale WVT for the ISEs in NC than for the ISEs in the YRV region. Hypothetically, this relatively strong (weak) water vapor pre-conditioning via large-scale WVT for the ISEs in NC (the YRV region) may be because that the water vapor for the precipitation in NC is mostly supplied by the large-scale WVT during the ISEs, whereas the water vapor for the precipitation in the YRV region might be largely supplied by the regional water vapor over the YRV region during the ISEs.

To examine the above hypothesis, the contribution of net water vapor flux and regional water vapor over the two target regions to the precipitation during time step −24 to time step 48 is computed using the method proposed by Sun and Wang [22], where the period from time step −24 to time step 0 is considered the water vapor pre-conditioning stage and the period from time step 0 to time step 48 is considered the intensive snowfall (IS) stage. The contribution of regional water vapor includes the contribution of original water vapor in the atmosphere and the contribution of regional evaporation to the total water vapor consumption for precipitation. Specifically, the relationship among precipitation, net water vapor flux and regional water vapor can be given by $\Delta P = \Delta WVT + \Delta O$ (1), where the $\Delta P$ represents the total water vapor consumption for precipitation, the $\Delta WVT$ represents the water vapor supplied by the simultaneous net water vapor flux induced by the large-scale WVT over the target regions and the $\Delta O$

represents the water vapor supplied by the original water vapor in the atmosphere over the target regions and by the regional evaporation over the target regions. The ΔP and ΔWVT can be computed based on the 12-h cumulative precipitation and 6-h net water vapor flux and the ΔO can be estimated using the above equation (1) given the information of ΔP and ΔWVT. It should be noted that the pre-conditioned water vapor over the target regions is considered a part of the original water vapor in the atmosphere over the target regions during the IS stage, hence the contribution of regional water vapor during the IS stage to the total precipitation may partially include the contribution of pre-conditioned water vapor.

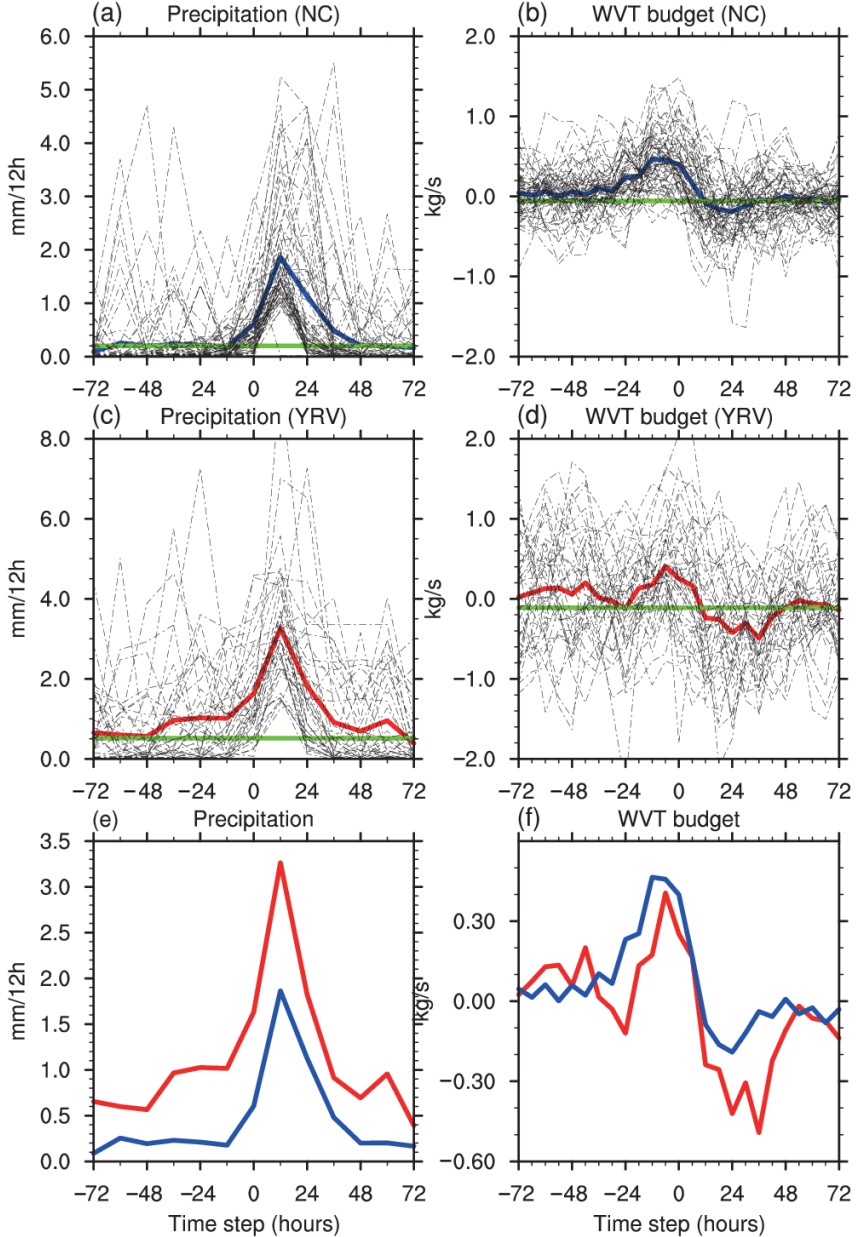

**Figure 6.** Evolution of (**a**,**c**) areal mean 12-h cumulative precipitation (mm) and (**b**,**d**) net water vapor flux ($10^8$ kg/s) over (**a**,**b**) NC and (**c**,**d**) the YRV region in the 72 h before and after the breakout of ISEs, where the black lines represent the individual cases, the thick blue and red lines represent the corresponding averages and the horizontal green lines represent the corresponding winter climatology. Evolution of (**e**) areal mean 12-h cumulative precipitation (mm) and (**f**) net water vapor flux ($10^8$ kg/s) over NC (in blue) and the YRV region (in red) in the 72 h before and after the breakout of ISEs. Time step 0 represents the beginning time of the ISEs.

As shown in Table 2, for the ISEs in NC, a large quantity of water vapor is pre-conditioned over NC via large-scale WVT during the water vapor pre-conditioning stage, which may account for 73% of the total water vapor consumption for precipitation during the ISEs in NC; whereas, for the ISEs in the YRV region, the pre-conditioned water vapor via large-scale WVT can only account for 16% of the total water vapor consumption during the ISEs in the YRV region. The simultaneous net water vapor flux during the IS stage is negative for the ISEs in both target regions, which is reflected by the negative values of contribution of ΔWVT (Table 2). On the other hand, for the ISEs in the YRV region, the regional water vapor supply during the IS stage may account for 112% of the total water vapor consumption for the precipitation during the ISEs (Table 2). Note that this result does not necessarily mean that the water vapor for the precipitation during the ISEs entirely comes from the regional water vapor but this result suggests that the regional water vapor potentially makes an important contribution to the ISE precipitation in the YRV region. The above results indicate an important influence of the large-scale WVT on the water vapor supply for the ISEs in NC and a less important influence of the large-scale WVT on the water vapor supply for the ISEs in the YRV region. In addition, the large contribution of regional water vapor to the ISEs in the YRV region (Table 2) may be a key reason for the relatively large ISE precipitation in the YRV region in comparison to that in NC (Figure 6e), because the precipitable water over the YRV region is generally notably larger than that over NC during winter (Figure 2c).

### 4.2. Synoptic Evolution of Cold Advection

Considering that the frequency and intensity of cold advection invading eastern China are closely associated with the SH [42,43], the synoptic evolution of SH during the ISEs in the NC and YRV regions are computed. As shown in Figure 7, an above-normal SH was observed during the ISEs in the NC and YRV regions (Figure 7a,b). In particular, the SH associated with the ISEs in the YRV region is notably stronger than the SH associated with the ISEs in NC in the 72 h before the breakout of ISEs and in the 24 h after the breakout of ISEs (Figure 7c). The above different strength of SH may result in different intensity of cold advection associated with the ISEs in the NC and YRV regions. As shown in Figure 8, in the 18 h before the breakout of ISEs in NC, the SLP anomalies over East Asia are characterized by negative SLP anomalies over central China and small SLP anomalies over Mongolia (the key region of SH), which induce southerly surface wind anomalies over NC (Figure 8a–d), resulting in little northerly cold advection invading NC. By contrast, as shown in Figure 9, in the 18 h before the breakout of ISEs in the YRV region, the SLP anomalies over East Asia are characterized by significant positive SLP anomalies over Mongolia and northern China, which may induce divergent surface wind anomalies and particularly northerly surface wind anomalies over the YRV region (Figure 9a–d), resulting in strong northerly cold advection invading the YRV region. The strong northerly cold advection invading the YRV region may play a key role in triggering the ISEs in the YRV region, which persists in the 24 h after the breakout of ISEs in the YRV region (Figure 9e–h).

Furthermore, to quantitatively compare the northerly cold advection associated with the ISEs in the NC and YRV regions, the synoptic evolution in the anomalies of areal mean surface meridional wind and surface temperature over NC and the YRV region are computed. As shown in Figure 10a, for the ISEs in NC, an increase in southerly wind anomaly is observed over NC in the 72–12 h before the breakout of ISEs and this southerly wind anomaly over NC changes towards a northerly anomaly after the breakout of ISEs; whereas, for the ISEs in the YRV region, an increasing northerly wind anomaly is observed over the YRV region in the 72 h before the breakout of ISEs in the YRV region, which reaches a maximum northerly wind anomaly approximating 2 m/s at time step 0, suggesting an important influence of the strong northerly cold advection on the breakout of ISEs. This strong cold advection over the YRV region gradually weakens after the breakout of ISEs. Correspondingly, the surface temperature over NC is above normal in the 72 h before the breakout of ISEs in NC (Figure 10b). Specifically, the surface temperature over NC has a positive anomaly of 0.88 °C at time step 0, indicating that there is no cold advection over NC when the ISEs are triggered. The temperature anomaly over NC falls towards a relatively small negative anomaly approximating −1 °C in the 48–72 h after the breakout

of ISEs in NC. In contrast, the surface temperature anomaly over the YRV region falls from a normal level at time step −24 towards a negative anomaly of −2.1 °C at time step 0 (Figure 10b), suggesting an impact of cold advection on the breakout of ISEs in the YRV region. The surface temperature anomaly over the YRV region further falls towards a significant negative anomaly around −5 °C in the 24–72 h after the breakout of ISEs, indicating a strong cold advection invading the YRV region.

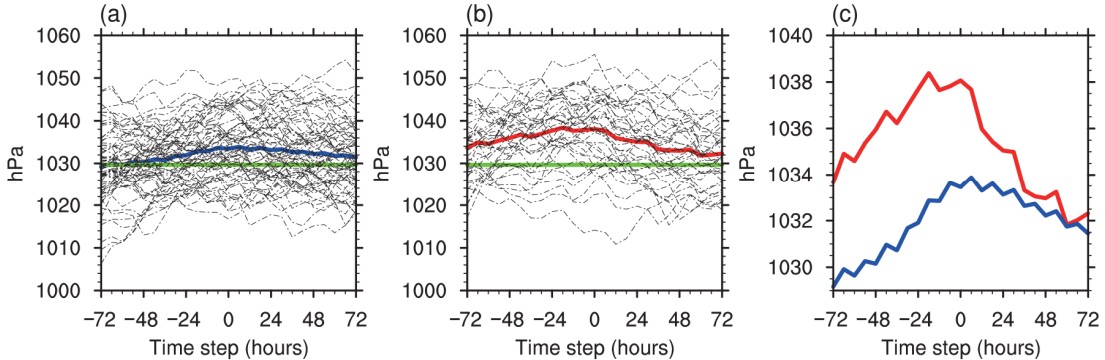

**Figure 7.** Evolutions of (**a**) the intensity of SH in the 72 h before and after ISEs occurred in NC and (**b**) the intensity of SH in the 72 h before and after ISEs occurred in the YRV region. Black lines represent the cases, the blue lines represent averages of SH during ISEs in NC, the red lines represent averages of SH during ISEs in the YRV region and the green horizontal lines represent the climatology of SH. Evolutions of (**c**) averages of the intensity of SH in the 72 h before and after ISEs occurred in NC (blue line) and in the YRV region (red line). Time step 0 represents the beginning time of the ISEs.

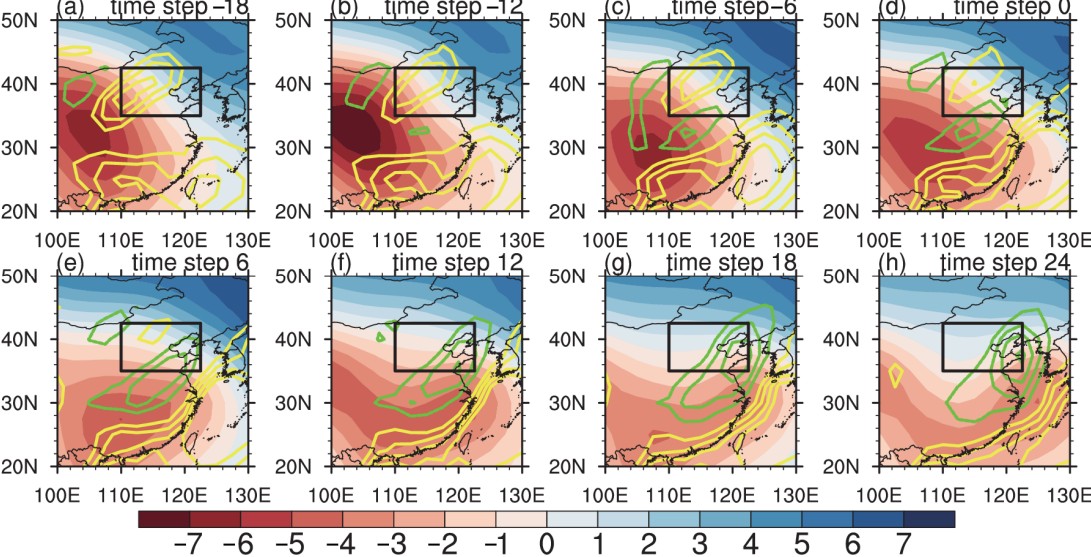

**Figure 8.** Evolution of SLP anomalies (colors, hPa) and surface meridional wind anomalies (contours, m/s; northerly wind anomalies are in green and southerly wind anomalies are in yellow) with respect to winter climatology at time steps (**a**) –18, (**b**) –12, (**c**) –6, (**d**) 0, (**e**) 6, (**f**) 12, (**g**) 18, and (**h**) 24 for the ISEs in NC.

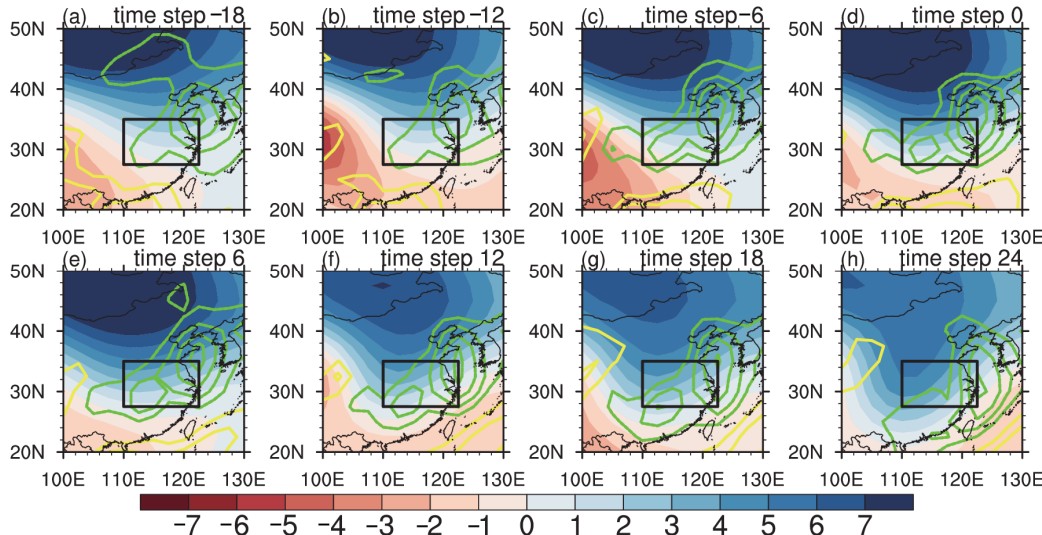

**Figure 9.** Evolution of SLP anomalies (colors, hPa) and surface meridional wind anomalies (contours, m/s; northerly wind anomalies are in green and southerly wind anomalies are in yellow) with respect to winter climatology at time steps (**a**) −18, (**b**) −12, (**c**) −6, (**d**) 0, (**e**) 6, (**f**) 12, (**g**) 18, and (**h**) 24 for the ISEs in the YRV region.

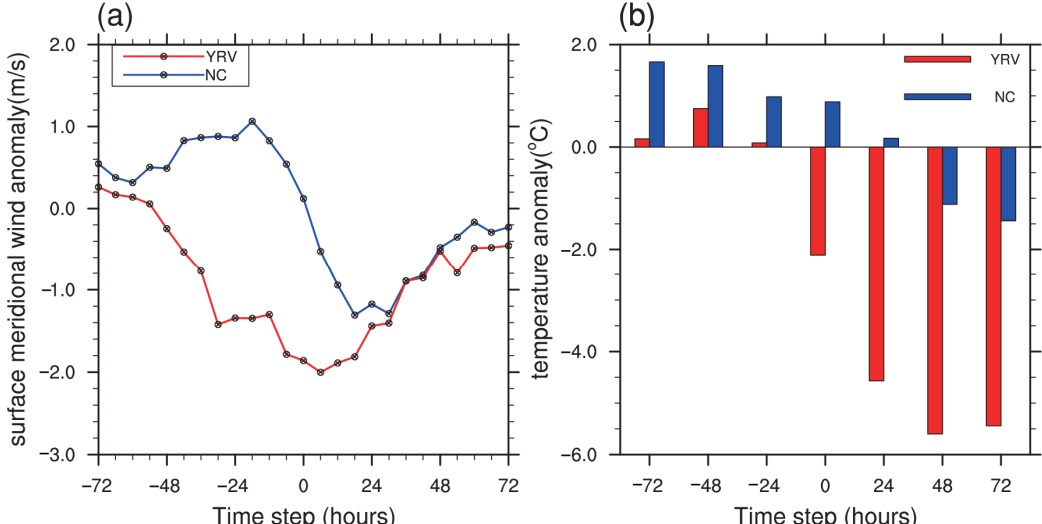

**Figure 10.** Evolution of (**a**) areal mean surface meridional wind anomalies (m/s) and (**b**) areal mean surface temperature anomalies (°C) over NC (in blue) and the YRV region (in red) in the 72 h before and after the breakout of ISEs.

The above results suggest an important role of strengthened SH and northerly cold advection for inducing the ISEs in the YRV region and a minor role of cold advection for inducing the ISEs in NC. It should be noted that the minor role of cold advection for inducing the ISEs in NC is mainly because that the ISEs in NC significantly depend on WVT rather than cold advection due to the dry winter climate in NC. The influence of cold advection on the ISEs in NC should not be ignored.

## 5. Conclusions and Discussions

Based on a composite analysis of 70 ISE cases in NC and 40 ISE cases in the YRV region, the synoptic evolution of WVT and cold advection associated with the ISEs in the NC and YRV regions are analyzed. The results indicate different roles of southerly WVT and northerly cold advection during the ISEs in the two regions. The ISEs in NC are closely associated with southerly WVT anomalies, which carry moisture from southern and eastern China into the NC region and the large-scale WVT

dominates the water vapor supply for the ISEs in NC. On the other hand, although the ISEs in the YRV region are also associated with southerly WVT anomalies over southern and eastern China, the large-scale WVT makes a relatively small contribution to the water vapor supply for the ISEs in the YRV region in comparison to that for the ISEs in NC, where the regional water vapor over the YRV region largely accounts for the water vapor supply for the ISEs in the YRV region. Regarding the role of cold advection, the ISEs in the YRV region are associated with a notably strengthened SH and a strong northerly cold advection invading the YRV region, whereas the ISEs in NC exhibit a less close relationship with northerly cold advection over NC. The above different roles of southerly WVT and northerly cold advection in the ISEs over the NC and YRV regions may be attributed to the different winter climate in the NC and YRV regions, where the southerly warm and moist air flow infrequently reaches NC during winter and the strong northerly cold advection infrequently reaches the YRV region during winter. Hence, the anomalous southerly WVT is a critical factor for triggering the ISEs in NC, whereas the northerly cold advection is a critical factor for triggering the ISEs in the YRV region.

The current study focuses on the common features in the synoptic evolution of WVT and northerly cold advection associated with the ISEs in the NC and YRV regions but does not go deep into the roles of southerly WVT and northerly cold advection for influencing the interannual variability of ISEs in the NC and YRV regions. Considering the different winter climate in the NC and YRV regions, the roles of WVT and cold advection in the interannual variability of ISEs in the NC and YRV regions might be different. In addition, it has been suggested that an anomalous Arctic sea ice may induce anomalous SH and hence impact the cold advections invading eastern China during winter and the tropical Pacific SSTs may induce anomalous WVT over eastern China during winter, whereby the Arctic sea ice and tropical Pacific SSTs are two important factors influencing the interannual variability of ISEs in eastern China [3,44]. Thus, the different influences of Arctic sea ice and tropical Pacific SSTs on the interannual variability of ISEs in the NC and YRV regions also deserve a further study.

Finally, the results of this study imply an importance of well-forecasted WVT to the forecast of ISE in NC as well as an importance of well-forecasted cold advection to the forecast of ISE in the YRV region, where a numerical forecast produced by a high-resolution and well-parameterized numerical model is demanded [45–47].

**Author Contributions:** Conceptualization, Z.X. and B.S.; Data curation, Z.X.; Formal analysis, Z.X. and B.S.; Funding acquisition, B.S.; Investigation, Z.X. and B.S.; Methodology, Z.X. and B.S.; Project administration, B.S.; Resources, B.S.; Software, Z.X.; Supervision, B.S.; Validation, Z.X. and B.S.; Visualization, Z.X.; Writing—original draft, Z.X. and B.S.; Writing—review & editing, B.S.

**Funding:** This study is funded by the National Key Research and Development Plan (Grant No. 2016YFA0600703), the Natural Science Foundation of China (Grant Nos. 41805047 and 41421004), the Natural Science Foundation of Jiangsu Province of China (Grant No. BK20180807), the Natural Science Foundation of Jiangsu Higher Education Institutions of China (Grant No. 18KJB170014), the Jiangsu Innovation and Entrepreneurship Team and the Priority Academic Program Development of Jiangsu Higher Education Institutions (PAPD) and the Scientific Research Foundation of Key Laboratory of Meteorological Disaster (KLME), Ministry of Education (KLME201803).

**Acknowledgments:** We sincerely thank the China Meteorological Administration and the National Centers for Environmental Prediction (NCEP) for providing the datasets used in this study. In addition, we sincerely thank the suggestions and help from Dapeng Zhang, Baoyan Zhu and Teng Wang.

**Conflicts of Interest:** The authors have no conflict of interest.

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
