# Peer review of "Different Roles of Water Vapor Transport and Cold Advection in the Intensive Snowfall Events over North China and the Yangtze River Valley"

_atmosphere, doi:10.3390/atmos10070368_

Round 1

Reviewer 1 Report

Review of Different roles of water vapor transport and cold advection in the intensive

snowfall events over North China and the Yangtze River valley by Zhixing Xie and Bo Sun to

be published in atmosphere

The topic of this paper was interesting. Intensive snowfall events can cause major damage to society

through injuries and hampering the traffic, as well as causing power cuts, damages to roof tops and

trees. The hypothesis of this paper was that southerly water vapor transport and northerly cold

advection are the triggering factors causing intensive snowfall events in the North China and

Yangtze River valley areas. This hypothesis is studied and confirmed with composite analysis. The

paper is well argued and have its place in intensive snowfall studies. The figures were good and

presented the results well. The only thing I felt is missing is the explanation why there is smaller

(ISE) precipitations in northern China than YRV region.

The language is mainly easy to read and there were no typos except in section 4.1 (I have marked

many, there might be more). There were lots of repeating same phrases inside one sentence which is

not that pleasant for the reader because the text does not flow nicely. Also the sentences were quite

long and could be cut to two or even three separate sentences in some cases.

My suggestion is that after spell check and minor revisions this paper is ready to be published in

atmosphere.

Minor comments:

Page 1.

Lines 20-25. Too long sentence. Please, cut to two. Suggestion: cut from line 23; '..,Suggesting..'

Line 36. Remove 'eastern China', already said in the beginning of sentence.

Page 2.

Line 15. replace 'eastern China and the Yellow Sea and the Japan Sea' with 'eastern China, the

Yellow Sea and the Japan Sea' and please check this kind of 'and and and'-list throughout the paper.

Page 4.

Table 2. Could precipitation amounts be added to table 2? Can the table be sorted according to the

date or the precipitation amount (if added) instead of time of the day?

Page 5.

Lines 19-20. '..,suggesting that northerly cold advection is a factor limiting the ISEs in the YRV

region.' What does this mean? Because there is no cold advection there is no ISEs OR because there

comes cold advection it prevents ISEs?

Page 6.

Figure 2a and 2c. Red text is hart to see from the figures.

Figure 2d. Could the dots be marked with larger contrast colors? Now they are quite similar.

Especially grey and blue, and orange and red.

Page 7.

Line 14. Replace 'southeast-northwest' with 'southwest-northeast'??

Lines 20-21. '..the southerly WVT may be a main limiting factor for the ISEs in

NC.' Why? Does the lack of WVT limit the ISEs in NC or why this is limiting factor?

Lines 24-25. '..whereby the northerly cold advection may be a main limiting factor for the ISEs in

the YRV region.' Why? Does the lack of northerly cold advection prevent the ISEs in YRV or why it

is the main limiting factor?

Lines 18-25. Please shorten the sentences. It is hard to follow long sentences with phrases

'for..which..whereas..whereby..'

Chapter 4.1 . Lots of typos. I have marked maybe most but there might still be some.

Lines 36-37. Replace eastern China and the South China Sea and western North Pacific conveyed''

with 'eastern China, the South China Sea and western North Pacific conveyed' and please check this

kind of 'and and and'-list throughout the paper.

Line 37. Remove 'into NC', already said in the beginning of sentence.

Line 39. Remove 'over NC', already said in the beginning of sentence. There is lots of repeating the

same phrases inside one sentence throughout the whole paper. Please reduce those.

Line 44. Replace 'YVR' with 'YRV'

Page 9.

Line 5. Replace 'synopitc' with 'synoptic' and 'precipitaion' with 'precipitation'

Line 7. Replace 'precipitaion' with 'precipitation'

Line 10. Replace 'bagan' with 'began'

Page 11.

Line 3. Replace 'incudes' with 'includes' ?

Line 5. Replace 'relationsihp' with 'relationship'

Line 17. Replace ''pre-condtioned' with 'pre-conditioned' ?

Line 21. Replace 'toal' with 'total' and 'simutaneous' with 'simultaneous'

Line 22. IS=Intensive snowfall?

Line 24. How something can account for 112% of total consumption?

Lines 26-29. This is hard to understand what is meant here. Please rephrase.

Table 2. Please add a small space between NC and YRV values to make the table clearer.

Page 13.

Line 18. '..there is little..' meaning what? That there is some but not enough cold advection? Or too

little? Or there is not enough cold advection? Or 'a little' / weak?

Page 14.

Lines 7-11. Please rephrase and shorten sentences, hard to understand.

Reviewer 2 Report

The present work analyzes intense snowfall events in two regions of China: North China and Yangtze River Valley. The analysis period is representative, since it comprises from 1951 to the present. Also the data handled are adequate: precipitation values registered in meteorological observatories, in addition to monthly values and subdaily, of surface pressure, surface wind, precipitable water vapor, and wind and specific humidity for different meteorological topographies. The analyzes carried out give consistency to the results and conclusions.

However, some considerations of minor importance are indicated:

1.- I think it is necessary to contextualize their study, carrying out a concise bibliographic review about important snowfall events at an international level and in the analyzed regions. In this sense, among other studies, the authors can consult: 1)Emilio Martínez-Ibarra, José Luis Serrano-Montes, J. Arias-García, Reconstruction and analysis of 1900–2017 snowfall events on the southeast coast of Spain, Climate Research, 2019…; 2.- Silvia Enzi, Chiara Bertolin, Nazzareno Diodato, Snowfall time-series reconstruction in Italy over the last 300 years, The Holocene, 2014…; 3) Nazzareno Diodato, Ulf Büntgen, Gianni Bellocchi. Mediterranean winter snowfall variability over the past millennium, International Journal of Climatology, 2018…

2.- Authors must specify the source of the data (the web link) of the data indicated in lines 1-2 on page 3.

Reviewer 3 Report

Review on “Different roles of water vapor transport and cold advection in the intensive snowfall events over North China and the Yangtze River valley

This study examines the influences of large-scale moisture transport and cold advection on the intensive snowfall events over two regions, i.e., one is the North China (NC), while the other is Yangtze River valley (YRV). They revealed that the intensification in Siberian High is more important over the southern region, i.e., YRV, which is a humid region, while the large-scale moisture supply is more important over the northern region, i.e., NC, which is a cold region. This is a well-written manuscript and the finding is interesting. My major worry about this manuscript is its introduction. The introduction goes to the questions without giving a good background. This manuscript may be published after clarifying the following points.

On the introduction:

Line 39 of page 1:  please cite this reference in place of that from Baidu, a reference tends to be more reliable than the information on a site

Ref: Zhou, B., et al. (2011), The great 2008 Chinese ice storm: Its socioeconomic–ecological impact and sustainability lessons learned, Bull. Am. Meteorol. Soc., 92(1), 47–60, doi:10.1175/2010BAMS2857.1.

Paragraph 2 (Lines 4-21 of page 2): Since the Siberian high is extremely important in your study, it is extremely important to describe it first in the head of this paragraph by including three aspects:

(1)   It is a dominant surface anticyclone over the Eurasian continent during winter (Ding, 1990);

(2)   It was found to be anti-correlated with the Arctic Oscillation (Gong et al. 2001).

(3)   However, Huang et al. (2016) has proved that its relationship with the Arctic Oscillation is of high non-linearity.

Ref:  Ding, Yihui. Meteorl. Atmos. Phys. (1990) 44: 281. https://doi.org/10.1007/BF01026822.

     Gong, D.-Y., S.-W. Wang and J.-H. Zhu (2001). "East Asian Winter Monsoon and Arctic Oscillation." 28(10): 2073-2076.

             Huang et al. (2016) On the Non-Stationary Relationship between the Siberian High and Arctic Oscillation. PLoS ONE 11(6): e0158122. https://doi.org/10.1371/journal.pone.0158122.

Paragraph 2 (Lines 4-21 of page 2): When the authors review the recent research progresses of the formation of intensive snowfall events, they mainly focused on the surface circulation and water vapor transport. It should be noted that the role of upper or middle tropospheric circulation anomalies also play an important role in triggering the precipitation extremes over eastern China with a long lead time. In particular, Huang et al. (2018) demonstrated that the formation of a Rossby wave train (namely the circumglobal teleconnection) along the subtropical jet stream favor the extreme precipitation over the southeastern Tibetan Plateau, while the dissipation of this wave train favors the precipitation extremes over southeastern China (Huang et al., 2019). Yang et al. (2019) further demonstrated that the coupling between the circumglobal global teleconnection and the Siberian blocking played a dominant role in driving the extreme snowfall events over eastern China. The authors should carefully review these processes in order to not mislead the readers that the extreme precipitation events during winter are a low-troposphere-only process.

Ref: Huang et al. (2018), On the formation mechanism for wintertime extreme precipitation events over the southeastern Tibetan Plateau, Journal of Geophysical Research: Atmospheres, 123(22), 12692–12714, https://doi.org/10.1029/2018JD028921.

Huang et al. (2019), A possible mechanism for the occurrence of wintertime extreme precipitation events over South China, Climate Dynamics, 52(3-4), 2367-2384, https://doi.org/10.1007/s00382-018-4262-8.

Yang et al. (2019), Synoptic conditions and moisture sources for extreme snowfall events over East China, Journal of Geophysical Research: Atmospheres, 124(2), 601–623, https://doi.org/10.1029/2018JD029280.

On Table 1: Why most of the intense snowfall events over YRV occur during the period 12:00 to 24:00 UTC? This is a very obvious phenomenon. The authors should discuss on it in the main text.

On Figure 3: From this figure, we could observe that the intense snowfall events over NC occur in tandem with that over the SC (3a-d), which means that it is hard to split the extreme precipitation events over NC and YRV into two different events. The authors should discuss in detail about this issue.

In addition, could the authors validate the rainfall and snowfall (and the total precipitation) over the two study regions of the NCEP-NCAR during the ISEs. The good performance of NCEP-NCAR in reproducing the precipitation could make the analyses in the whole manuscript more solid and acceptable by the readers.
